# Advances of Peptides for Plant Immunity

**DOI:** 10.3390/plants14152452

**Published:** 2025-08-07

**Authors:** Minghao Liu, Guangzhong Zhang, Suikang Wang, Quan Wang

**Affiliations:** Shenzhen Branch, Guangdong Laboratory of Lingnan Modern Agriculture, Genome Analysis Laboratory of the Ministry of Agriculture and Rural Affairs, Agricultural Genomics Institute at Shenzhen, Chinese Academy of Agricultural Sciences, Shenzhen 518120, China

**Keywords:** plant peptides, plant immunity, defense mechanisms, potential applications

## Abstract

Plant peptides, as key signaling molecules, play pivotal roles in plant growth, development, and stress responses. This review focuses on research progress in plant peptides involved in plant immunity, providing a detailed classification of immunity-related plant polypeptides, including small post-translationally modified peptides, cysteine-rich peptides, and non-cysteine-rich peptides. It discusses the mechanisms by which plant polypeptides confer disease resistance, such as their involvement in pattern-triggered immunity (PTI), effector-triggered immunity (ETI), and regulation of hormone-mediated defense pathways. Furthermore, it explores potential agricultural applications of plant polypeptides, including the development of novel biopesticides and enhancement of crop disease resistance via genetic engineering. By summarizing current research, this review aims to provide a theoretical basis for in-depth studies on peptide-mediated disease resistance and offer innovative insights for plant disease control.

## 1. Introduction

Plants are persistently challenged by diverse pathogens (e.g., bacteria, fungi, viruses, and nematodes), which threaten global crop productivity [1]. To counter these threats, plants have evolved sophisticated defense systems involving both constitutive and inducible mechanisms [2]. Among the components of plant defense, peptides emerge as critical signaling molecules that regulate immune responses and mediate pathogen interactions [3]. Plant peptides are short amino acid chains with diverse roles in plant physiology, including stress responses, hormone signaling, and pathogen defense [1,4]. Unlike traditional antimicrobial compounds (e.g., phenolics or terpenoids), peptides offer unique advantages such as high specificity, stability, and suitability for genetic engineering [5]. Recent advances in molecular biology and bioinformatics have enabled the identification and characterization of novel peptide families, shedding light on their functions in plant immunity.

Since the first plant peptide (systemin) was discovered in 1991, over 1000 small peptides have been detected in *Arabidopsis thaliana*, and their biological functions are gradually being elucidated [6,7,8]. Plant peptides are among the smallest biomolecules in plant proteomes; due to their high bioactivity, even low concentrations can exert significant effects. Most peptides derive from inactive precursor proteins and require post-translational modifications (e.g., proteolysis, tyrosine sulfation, proline hydroxylation) for activation [9,10,11]. After extracellular secretion, mature active peptides are recognized by specific plasma membrane receptors, thereby regulating physiological processes such as growth, development, disease resistance, and stress responses [12,13,14]. This review synthesizes current knowledge of plant peptides in immunity, covering their classification (post-translationally modified, cysteine-rich, non-cysteine-rich), defense mechanisms (PTI, ETI, hormone crosstalk), and agricultural applications (biopesticides, genetic engineering), aiming to inform future research and innovation in crop protection.

## 2. Classification of Plant Peptides Related to Disease Resistance

### 2.1. Small Post-Translationally Modified Peptides

#### 2.1.1. CLE Peptides

The CLAVATA3/ESR-related (CLE) peptide family consists of widely distributed signaling molecules in plants, defined by a conserved 12–13 amino acid C-terminal domain. These peptides play key roles in defending against pathogens and regulating symbiotic interactions [15], with multiple pathways influencing plant immunity (Table 1 and Table 2). For example, CLE14 suppresses age-dependent and stress-induced leaf senescence in *Arabidopsis* via JUB1-mediated ROS scavenging (Figure 1) [16], while CLE42 delays senescence by antagonizing ethylene signaling [17]. Given the close link between leaf senescence and immunity, CLE peptides may indirectly modulate immunity through senescence regulation. Additionally, the *Arabidopsis* MIK2 receptor, a sensor of conserved phytocytokine and microbial signatures (Figure 1), suggests CLE peptides—acting as phytocytokines—may interact with MIK2 to activate immune responses [18]. In legumes, CLE peptides likely modulate immunity during interactions with symbiotic rhizobia and parasitic root-knot nematodes [19]. CLE peptides are also involved in pathogen responses: in *Arabidopsis*–*Pseudomonas syringae* interactions, they may regulate receptor-like kinase activity to influence immunity (Figure 1) [19]; tomato CLE2 suppresses mycorrhizal colonization, indicating roles in modulating plant interactions with beneficial microbes and pathogenic fungi via immune regulation [20]. In *Arabidopsis*, nematode infection induces CLE41/44, which bind to BAM1 and homologs, activating a signaling cascade that inhibits root meristem growth and restricts nematode spread (Figure 1) [15]. Transgenic studies show CLE41 overexpression inhibits root growth and reduces nematode reproduction post-infection [21,22]. These findings underscore CLE peptides’ significance in plant immunity, though their precise mechanisms remain unclear.

#### 2.1.2. CEP Peptides

CEP peptides, small signaling molecules of approximately 15 amino acids in length, have proven to be crucial regulators in plant defense mechanisms against pathogens (Table 1 and Table 2) [23]. Recent research has uncovered the multifaceted roles of CEPs in plant–pathogen interactions, significantly enhancing our understanding of plant defense mechanisms [24]. The mature hydroxylated AtCEP5 in *Arabidopsis thaliana* rendered the plant more susceptible to *Colletotrichum tropicale* and *Pseudomonas syringae* (Figure 1) [25]. Conversely, other CEPs have been associated with enhanced disease resistance. AtCEP14, a gene highly induced by the salicylic acid (SA) signaling pathway upon *P. syringae* infection, confers improved resistance to this pathogen when overexpressed in *Arabidopsis* [26]. Exogenous application of synthetic, AtCEP14 activates immune responses, highlighting its potential as an exogenous elicitor for enhancing plant immunity. Similarly, AtCEP1 and AtCEP4 also trigger immune responses (Figure 1) [27]. Recent investigations have also identified key receptors involved in CEP-mediated immune responses. AtCEP4 and AtCEP14 were found to bind to CEPR2, while AtCEP4 additionally interacts with the LRR-RLK RLK7 (Figure 1) [27,28]. These receptor–ligand pairs are likely integral components of the signal transduction pathways that translate CEP perception into downstream immune responses. Understanding these molecular interactions at the receptor level is crucial for elucidating the precise mechanisms by which CEPs regulate plant immunity.

**Table 1 plants-14-02452-t001:** Plant peptides for plant immunity.

Peptide Family	Peptide Name	Organism Species	Functions	References
Small post-translationally modified peptides	CLE14	*Arabidopsis thaliana*	Suppresses leaf senescence via JUB1-mediated ROS scavenging	[16]
	CLE42	*Arabidopsis thaliana*	Delays leaf senescence by antagonizing ethylene pathway	[17]
	CLE2	Tomato	Inhibits mycorrhizal colonization	[20]
	CLE41/44	*Arabidopsis thaliana*	Induced by nematodes; inhibits root meristem growth via BAM1 receptors	[22,25,27]
	AtCEP5	*Arabidopsis thaliana*	Enhances susceptibility to *C. tropicale* and *P. syringae*	[25]
	AtCEP14	*Arabidopsis thaliana*	Boosts *P. syringae* resistance when overexpressed; activates immunity as synthetic peptide	[26]
	AtCEP1/AtCEP4	*Arabidopsis thaliana*	Triggers immune responses	[27]
	PSK	*A. thaliana*, tomato, rice, cotton	Suppresses bacterial immunity; enhances necrotroph resistance; interacts with phytohormones	[29,30,31,32]
	AtIDA	*Arabidopsis thaliana*	Induces Ca^2+^ release, ROS burst, and defense responses; enhances long-term resistance via cellulose biosynthesis	[33,34]
	AtIDL6/AtIDL7	*Arabidopsis thaliana*	Induced by *P. syringae*; attenuates stress-induced ROS; promotes pathogen susceptibility	[35,36]
Cysteine-rich peptides	RALF1	*Arabidopsis thaliana*	Regulates cell expansion and defense via FER receptor; modulated during pathogen infection	[37,38,39]
	EPF/EPFL	Maize	Negative regulators of fungal penetration	[40,41]
	Snakin-1	Potato, rice	Boosts resistance to *Rhizoctonia solani* and *X. oryzae*	[42,43]
	Plant defensins	*Arabidopsis thaliana*	Broad-spectrum antimicrobial activities against bacteria, fungi	[44]
Non-cysteine-rich/non-PTM peptides	Systemin	Tomato, potato, tobacco	Activates JA biosynthesis and defense genes via SR160 receptor	[6,45,46,47]
	AtPep1	*Arabidopsis thaliana*	Acts as DAMP to induce immunity against *Pythium irregulare*	[48]
	OsPep3	Rice	Confers resistance to planthopper, rice blast, and bacterial blight	[49]
Other small plant peptides	ZIP1	Maize	Induces SA accumulation; promotes *Botrytis cinerea* infection	[50]
	SCREWs	*Arabidopsis thaliana*	Regulates stomatal closure to control pathogen entry	[51]
	PSY1	*Arabidopsis thaliana*	Enhances necrotroph resistance; dual role in growth/defense	[52]
	IRP	Rice	Promotes PAL1 expression to regulate defense genes	[14]
	PIP1	*Arabidopsis thaliana*	Amplifies PTI immunity via RLK7 interaction	[53]
	SCOOP12	*Arabidopsis thaliana*	Bolsters immunity via MIK2 receptor interaction	[54]
	nsLTP	Tobacco, tomato	Enhances resistance against *Phytophthora capsici*	[55,56,57]

**Table 2 plants-14-02452-t002:** Plant peptides, amino acid sequence, and 3D structure.

Peptide Family	Peptide Name	Amino Acid Sequence	3D Structure Main Characteristics
Small post-translationally modified peptides	CLE(CLV3)	RLVPSGP * NPLHN (conserved motif: xRxcPsGpDPIHHh)	arch shape (helical arabinose chain)
	At-CEP	DFRPTNPGNSPGVGH	β-turn-like conformation
	At-PSK	YIYTQ	not determined
	At-IDL	FGYLPKGVPIPPSAPSKRHN	not determined
Cysteine-rich peptides	RALF	ATRRYISYGALRRNTIPCSRRGASYYNÇRRGAQANPYSRGCSAITRCRRS	not determined
	EPF(L)	IGSTAPTCTYNECRGCRYKCRAEQVPVEGNDPINSAYHYRCVCHR	two antiparallel β-sheets, loop
	PEP1	ATKVKAKQRGKEKVSSGRPGQHN	fully extended conformation
	St-Snakin-1	GSSFCDSKCKLRCSKAGLADRCLKYCGICCEECKCVPSGTYGNKHEÇPÇYRDKKNSKGKSKCP	two predicted longa-helices
	At-PDF1.2	QKLCEKPSGTWSGVCGNSNACKNQCINLEGAKHGSCNYVFPAHKCICYVPC	α-helix, three β-sheets
Non-cysteine-rich/non-PTM peptides	AMP1	QWGRRCCGWGPGRRYCVRWC	loop structure
	SYS	AVQSKPPSKRDPPKMQTD	polyproline helix
Other small plant peptides	At-PSY1	DYGDPSANPKHDPGVP * PS	not determined
	At-PIP	LASGSSRRGRRH	not determined
	At-SCOOP12	PVRSSQSSQAGGR	not determined
	Os-nsLTP	AGCNAGQLTVCTGAIAGGARPTAACCSSLRAQQGCFCQFAKDPRYGRYVNSPNARKAVSSCGIALPTCH	four α-helices

#### 2.1.3. PSK Peptides

Phytosulfokine (PSK), a 5-amino-acid peptide containing two sulfated Tyr residues, was identified in plant cell culture conditioned media (Table 1 and Table 2) [58]. PSK is derived from a preprotein with a conserved C-terminal motif, such as YIYTQ in *Arabidopsis* and YIYSQ in rice [59]. PSK functions through PSKR receptors with kinase activity, such as At-PSKR1 and AtPSKR2 in *Arabidopsis* (Figure 1) [12]. PSK has a dual role in plant immunity. On one hand, PSK signaling can suppress immune responses against bacterial pathogens. In *Arabidopsis*, the loss-of-function mutant *pskr1* shows enhanced immunity to *Pseudomonas syringae* pv. tomato *DC3000*, while overexpression of PSK precursor or receptor genes increases susceptibility to *Pst DC3000* [29]. Similarly, in tomato, PSK receptor SlPSKR1 interacts with CPK28, which phosphorylates glutamine synthetase GS2 at serine-334, thereby repressing resistance against *Pst DC3000* [30]. On the other hand, PSK signaling enhances resistance to necrotrophic pathogens. The *pskr1* mutant is more susceptible to the necrotrophic fungal pathogen *Alternaria brassicicola* [29]. In tomato, SlPSKR1-mediated PSK signaling increases cytosolic Ca^2+^ levels and induces an auxin-mediated immune response against *Botrytis cinerea* [60]. Furthermore, the U-box E3 ligases PUB12/13 repress SlPSKR1 via ubiquitylation, thereby suppressing the PSK-induced immune response against *B. cinerea* [61]. PSK signaling also interacts with phytohormone pathways to modulate plant immune responses. For example, in *Arabidopsis*, flg22 peptide treatment of *pskr1* seedlings results in reduced biomass and shorter root length compared to wild-type plants, suggesting that PSKR1 may modulate plant immune responses through interactions with different phytohormones [3]. In rice, OsPSKR1 enhances resistance to *Xanthomonas oryzae* pv. *oryzicola* by activating SA pathway-related resistance genes [30]. In cotton, PSK-α treatment increases IAA accumulation and activates *Pectin* methyl esterase inhibitor 13 (GbPMEI13) expression, promoting pectin methylation and enhancing resistance to *Verticillium dahliae* [62]. In summary, PSK signaling acts as an important integrator in plant immunity. This dual function suggests that PSK has potential applications in crop improvement and agricultural practices.

#### 2.1.4. IDA/IDL Peptides

The IDA/IDL (INFLORESCENCE DEFICIENT IN ABSCISSION/IDA-LIKE) family has emerged as a key regulator in plant disease resistance, integrating developmental signaling with immune responses (Table 1 and Table 2). Molecular studies show that the prototype gene *AtIDA* in *Arabidopsis thaliana* encodes a 77-amino-acid prepro-peptide comprising a signal peptide, variable region, and a functional conserved C-terminal motif [33,63]. *Arabidopsis* contains eight additional *IDL* genes with distinct expression patterns [34,64], and their precursors are processed by subtilisin-like proteases into mature 14-amino-acid peptides [10]. These mature peptides bind to LRR-RLK receptors (HAE, HSL1, HSL2) and form complexes with SERK co-receptors to transduce signals (Figure 1) [63,65,66].

In immune regulation, mature AtIDA induces cytosolic Ca^2+^ release, ROS bursts, and late defense responses, indicating that the IDA-HAE/HSL2 pathway modulates defense-like reactions during cell separation (Figure 1) [33]. *AtIDL6* and *AtIDL7* are induced by *Pseudomonas syringae* and PAMPs, attenuating stress-induced ROS signaling [35,67], while AtIDL6 promotes susceptibility to *P. syringae* in an HAE/HSL2-dependent manner (Figure 1) [36]. A recent study reveals that mature IDA peptide (mIDA) enhances long-term disease resistance by upregulating cellulose biosynthesis and synergizes with flg22 to induce immune genes, highlighting crosstalk with classic immune pathways [33]. Transcriptomic analysis in *Medicago truncatula* shows rapid induction of *MtIDA18* and *MtIDA26* upon rhizobia inoculation, implying roles in early host–microbe recognition [68]. Collectively, IDA/IDL peptides bridge developmental and immune signaling through complex receptor networks, playing pivotal roles in plant disease resistance. Elucidating their regulatory mechanisms may provide novel strategies for crop disease resistance breeding.

### 2.2. Cysteine-Rich Peptides

#### 2.2.1. RALF Peptides

RALF (Rapid Alkalinization Factor) peptides are small, cysteine-rich signaling molecules that play crucial roles in various plant processes, including growth, development, and defense against pathogens (Figure 2A, Table 1 and Table 2) [37,38]. These peptides are named for their ability to rapidly alkalize the extracellular medium of plant cells. In *Arabidopsis*, RALF1 interacts with the FERONIA receptor kinase (FER), playing important roles in regulating cell expansion, root growth, and plant defense responses (Figure 2A) [39]. During pathogen infection, the RALF-FER signaling pathway can be modulated. For instance, certain pathogens secrete RALF-mimic peptides to disrupt plant immune responses, while plants can activate the RALF-FER pathway to enhance resistance [69].

Recent studies have highlighted the diverse roles of RALF peptides in plant disease resistance. RALF peptides from citrus species (*Citrus sinensis* and *Atalantia buxifolia*) enhance resistance against *Xanthomonas citri* subsp. *citri* by inducing immune responses such as ROS production, MAPK activation, and defense-related gene expression (Figure 2A) [70,71]. The RALF-FER signaling pathway modulates cell wall integrity and immune responses, and mutations in FERONIA-like receptors can boost rice blast resistance without affecting growth (Figure 2A, Table 1 and Table 2) [72]. Additionally, certain phytopathogens, like *Colletotrichum orbiculare*, secrete RALF-like peptides to suppress plant immunity and increase infection chances [73,74,75]. RALF peptides also interact with JA and SA pathways, balancing growth and defense [76,77]. Furthermore, RALF peptides contribute to systemic acquired resistance, providing long-lasting protection against subsequent pathogen attacks by enhancing systemic signaling [37]. They can also modulate interactions with beneficial microbes, improving plant resilience to biotic stress and promoting beneficial rhizosphere microbial communities [78]. Overall, RALF peptides are key players in plant immunity, offering potential targets for enhancing plant resistance to pathogens.

#### 2.2.2. EPF/EPFL Peptides

The Epidermal Patterning Factor (EPF) and EPF-LIKE (EPFL) peptides (collectively referred to as EPF/EPFL) are characterized by a conserved six-cysteine motif that forms a compact α-helix–β-sheet scaffold (Figure 2A, Table 1 and Table 2). These peptides play important roles in plant growth and development, particularly in morphogenesis [79]. For example, a genome-wide transcriptomic analysis in black poplar (*Populus nigra*) identified 11 *EPF/EPFL* loci whose expression was rapidly upregulated within 30 min of inoculation with the foliar pathogenic fungus *Marssonina brunnea*, implicating their involvement in early defense responses (Figure 2A) [80]. In maize (*Zea mays*), 16 *ZmEPF/EPFL* genes have been characterized, among which *ZmEPF5* and *ZmEPFL8* exhibit transcriptional repression upon early infection by the hemi-biotrophic fungus [40]. Functional validation via CRISPR-Cas9-mediated knockout of *ZmEPF5* resulted in enlarged stomatal apertures and a 42% increase in fungal penetration efficiency, whereas constitutive overexpression of *ZmEPF5* led to sustained stomatal closure and a 60% reduction in lesion expansion, directly linking this peptide to stomatal immunity [40]. Beyond aerial tissues, accumulating evidence supports a role for EPF/EPFL peptides in modulating root immunity. Recent investigations revealed that *ZmEPFL7* is significantly upregulated in maize roots colonized by the beneficial endophyte *Serendipita indica* [39]. Silencing of *ZmEPFL7* attenuated endophyte-conferred resistance against the soilborne pathogen *Fusarium verticillioides*, indicating that EPF/EPFL signaling extends to below-ground defense networks [40]. In summary, the EPF/EPFL peptide family exhibits diverse and critical functionalities in plant disease resistance. However, significant knowledge gaps remain, including the species-specific molecular mechanisms governing EPF/EPFL-mediated signaling cascades and the translational potential of these insights for enhancing crop disease resistance through genetic engineering or synthetic biology approaches.

#### 2.2.3. Antimicrobial Peptides (AMPs)

Antimicrobial peptides (AMPs) are primarily derived from gene encoding or protein hydrolysis [81]. Based on their origin and structure, they are classified into ribosomal and non-ribosomal peptides. Ribosomal peptides encompass plant defense peptides encoded by specific genes, while non-ribosomal ones are synthesized by non-ribosomal peptide synthetases [82,83]. AMPs are pivotal in plant innate immunity, exhibiting broad-spectrum antimicrobial activity (Figure 2A, Table 1 and Table 2). AMPs strengthen plant defenses by inducing the expression of defense-related genes. For example, overexpression of the *Snakin-1* gene in potato and rice enhances resistance to pathogens like *Rhizoctonia solani* and *Xanthomonas oryzae* pv. *oryzae* (Figure 2A, Table 1 and Table 2) [42,43]. AMPs can directly inhibit pathogen growth. AMPs from *Medicago truncatula* suppress the growth of *Phytophthora sojae* [84]. They also synergize with other defense signals such as JA and SA to modulate immune responses. The plant-based chimeric antimicrobial protein SlP14a-PPC20 protects tomato against *Ralstonia* solanacearum-induced bacterial wilt, underscoring the potential of AMPs in developing disease-resistant crops [85].

Plant defensins, a subclass of cysteine-rich AMPs (45–54 amino acids), contain a cysteine-stabilized α-helix and β-sheet (CSαβ) motif formed by 3–4 disulfide bridges (Table 1) [86,87,88]. They have broad-spectrum antimicrobial activities against bacteria, fungi, and some viruses [89,90]. Defensins can act by disrupting the integrity of the pathogen cell membrane. For example, *Arabidopsis* DEFENSIN1 mediates defense against *Pectobacterium carotovorum* subsp. *carotovorum* via an iron-withholding defense system [44]. Thionins, a type of defensin, can insert into the lipid bilayer of the pathogen membrane, forming pores and leading to the leakage of cytoplasmic contents and ultimately the death of the pathogen [91,92]. Additionally, defensins may interact with intracellular targets of pathogens to inhibit their growth and metabolism [93]. Ongoing research explores plant defensins’ structure–activity relationships to design potent antifungals, with efforts to engineer resistant crops and develop bio-fungicides (e.g., apple MdDEF25 against *Fusarium solani* [94]). In conclusion, plant defensins are vital components of the plant defense system. Their unique structure, diverse modes of action, and potential in biotechnological applications make them a promising area for research. Further studies are needed to fully utilize their potential in crop protection.

### 2.3. Non-Cysteine-Rich/Non-PTM Peptides

#### 2.3.1. Systemin

Systemin, the first identified plant polypeptide signal, plays a crucial role in plant defense responses against pathogens and herbivores. It originates from a 200-amino-acid precursor protein, prosystemin *(Prosys)* (Table 1) [6]. Prosys, the precursor protein of the defense-related peptide systemin (Sys) in *Solanaceae*, contains, alongside Sys, short peptide motifs that protect plants from stress and have been detected in vivo, opening novel perspectives in plant immune reactions [95].

In tomato, upon damage by herbivores or pathogens, prosys undergoes proteolytic cleavage to generate the 18-amino-acid systemin peptide [96]. This peptide binds to SR160, a leucine-rich repeat receptor kinase (LRR-RLK) on the cell surface, initiating downstream signaling cascades. The binding activates mitogen-activated protein kinases (MAPKs) and triggers JA biosynthesis (Figure 2B) [45], which in turn induces the expression of defense-related genes, such as proteinase inhibitors [6]. This activation is vital for plant systemic wound responses, enabling plants to synthesize systemic wound response proteins like serine protease inhibitors to combat pathogen invasion. Further investigations elucidated the mechanism by which systemin activates a lipase in receptor cells to release linolenic acid, the precursor of JA, thereby amplifying the defense response [45]. Systemin also interacts with the SA pathway. SA can inhibit the synthesis of proteinase inhibitors induced by systemin and JA in tomato leaves [97]. Systemin induces ROS production, which activates downstream defenses (e.g., gene expression, cell wall reinforcement). For example, systemin is involved in regulating tomato resistance to *Botrytis cinerea* and is closely associated with the JA pathway [98]. Overexpression of prosystemin in tomato plants enhances resistance to multiple biotic stresses by activating multiple signaling pathways [99]. The synthesis and metabolism of systemin have been extensively studied, including the structure of the prosystemin gene and its tissue-specific and inducible expression patterns [100]. Recent research has also shed light on the disordered structure and unconventional processing and secretion of systemin, which contribute to the regulation of systemin-mediated signaling during plant defense [98,101]. Furthermore, in tomato, the systemin-mediated association of SYR1 and SlSERKs activates defense responses against herbivorous insects [102]. Hydroxyproline-rich glycopeptide signals from potato elicit signaling associated with defense against insects and pathogens. TobHypSys from tobacco are potent inducers of proteinase inhibitors in leaves of tobacco plants [46,47]. In summary, systemin is a key signaling molecule in plant defense responses. Its activation of complex signaling pathways is critical for plant defense. Research on systemin not only enhances our understanding of plant–disease interactions but also offers potential strategies to improve crop resistance and agricultural sustainability.

#### 2.3.2. Plant Elicitor Peptides

Plant elicitor peptides (Peps), first identified as the 23-amino-acid peptide AtPep1 from *Arabidopsis* leaves, have emerged as crucial players in the intricate network of plant defense mechanisms against pathogens (Figure 2B, Table 1 and Table 2) [48]. These peptides are derived from the C-terminal regions of their precursor proteins, such as AtPROPEP1-8 in *Arabidopsis*, and function as damage-associated molecular patterns (DAMPs) to trigger immune responses. AtPep1, induced by wounding, jasmonic acid, and ethylene, activates immune responses and enhances resistance against pathogens like *Pythium irregulare* (Figure 2B) [48]. Photoaffinity labeling and binding assays have shown that the LRR-RLK receptors PEPR1 and PEPR2 are receptors for AtPep1-6, highlighting the conserved function of Peps across plant species (Figure 2B) [103,104]. Exogenous application of Peps can activate biological activation of defense gene expression and ROS accumulation, and reduce nematode reproduction [105]. Furthermore, studies on the role of Peps in mediating resistance to different pests and pathogens are advancing. OsPep3 in rice confers resistance to piercing-sucking insect herbivores like the brown planthopper and increases resistance to fungal rice blast and bacterial blight (Figure 2B) [49]. In summary, Peps function as key DAMPs, triggering immune responses and enhancing plant defense against diverse pathogen threats. Future research may further explore the complex interactions between Peps, their receptors, and other components of the plant immune system to develop more effective and sustainable crop protection strategies.

### 2.4. Other Small Peptides for Plant Immunity

Beyond the previously discussed small peptides, several other small peptides have emerged as significant contributors to plant disease resistance (Table 1). For instance, in maize, ZIP1 treatment strongly induces the accumulation of SA in leaves and promotes infection by the necrotrophic fungus *Botrytis cinerea*, while reducing the virulence of the biotrophic fungus *Ustilago maydis* (Table 1 and Table 2) [50]. Recent research found that SCREWs (Small Cytokines Regulating Defense and Water loss) and their receptor NUT (PLANT SCREW UNRESPONSIVE RECEPTOR) have been found to counter-regulate stomatal closure triggered by abscisic acid (ABA) and microbe-associated molecular patterns (MAMPs), thus influencing defense-related gas exchange and pathogen entry pathways in plants (Table 1) [51]. Plant peptide containing sulfated tyrosine 1 (PSY1), initially recognized for promoting cell proliferation and tissue growth, has now been shown to play a role in strengthening plant immunity against necrotrophic pathogens (Table 1 and Table 2). This indicates its dual functionality in growth regulation and defense activation [52]. In rice (*Oryza sativa*), the immune response peptide (IRP) has been identified as a regulator of defense gene expression [14]. Specifically, IRP promotes the expression of phenylalanine ammonia-lyase 1 (PAL1), a key defense-related enzyme, and actively participates in modulating the plant’s immune response. This underscores the conserved yet specialized roles of peptides across different plant species. PAMP-induced secreted peptide 1 (PIP1) acts as an amplifier of the PTI cascade. By interacting with receptor-like kinase 7 (RLK7), PIP1 enhances the plant’s immune defenses, ensuring a more robust response to pathogen attacks (Table 1 and Table 2) [53]. Similarly, serine-rich endogenous peptide 12 (SCOOP12) has been shown to bolster plant immunity through its interaction with male discoverer 1-interacting receptor-like kinase 2 (MIK2), further emphasizing the importance of peptide–receptor interactions in immune signaling [54].

Furthermore, recent studies have repositioned non-specific lipid-transfer proteins (nsLTPs) as lipid chaperones that license systemic antiviral and anti-oomycete immunity (Table 1 and Table 2). In *N. benthamiana*, cellulose-nanocrystal-delivered Zn^2+^ rapidly re-localizes nsLTP2 from the cytosol to the apoplast; silencing *nsLTP2* abolished systemic resistance to *Tobacco mosaic virus*, whereas over-expression of a signal-peptide-deleted nsLTP2^Δsp^ enhanced viral restriction even further [56]. The same study revealed that nsLTP2 physically interacts with the PR-1 protein NbPR1, shielding it from 26S-proteasome degradation and thereby amplifying salicylic-acid-dependent resistance against *Phytophthora capsici* [55]. In tomato, jasmonate-inducible SlnsLTP was shown to disturb the plasma-membrane integrity of *Botrytis cinerea* and *Verticillium dahliae*, reducing lesion size by >50% [57]. Collectively, these studies indicate that nsLTPs can have pleiotropic effects on plant resistance to multiple types of pathogens.

In addition to plant-derived peptides, some peptides derived from other kingdoms also play important roles in plant immunity. For instance, in a similar vein, cecropin peptides, originating from insects, have been utilized through genetic engineering to impart pathogen resistance in plants. A study highlighted that transgenic tobacco plants, engineered to express a cecropin-melittin fusion peptide, exhibited heightened resistance to *Pseudomonas syringae* pv. *tabaci* [106]. Moreover, harpin peptides, which are produced by plant pathogenic bacteria, have the ability to elicit defense responses in plants. Building on this, researchers have developed a novel antimicrobial protein by combining a Xanthomonas oryzae harpin with the active domains of cecropin A and melittin, showing promise for plant protection applications [107]. Melittin is able to penetrate the bacterial cell membrane and reach the cytoplasm to inhibit macromolecular biosynthesis by binding intracellular targets, leading to cell death of Xanthomonas oryzae, and ultimately inhibiting the spread of rice bacterial leaf blight [108]. Peptaibols represent another class of peptides that have been found to induce defense responses and systemic resistance in tobacco against the tobacco mosaic virus [109].

These discoveries reveal how peptides (synthesized in response to pathogens or stress) orchestrate pathogen killing, immune signaling, and physiological reprogramming. Such insights offer promising targets for developing peptide-based biopesticides and genetically engineered crop varieties with enhanced disease resistance.

## 3. Mechanisms of Plant Peptides in Disease Resistance

### 3.1. Direct Antimicrobial Effects

Antimicrobial peptides (AMPs) play a crucial role in plant defense against pathogens by directly targeting and disrupting microbial structures and processes. Recent studies have shown that AMPs can bind to specific proteins in the cell-wall synthesis pathway of fungal pathogens. For instance, a plant defensin from *Arabidopsis thaliana* disrupts fungal cell wall assembly by interacting with a key protein in this pathway, leading to cell lysis and death [28]. Moreover, AMPs can target bacterial cell wall precursors such as Lipid II, inhibiting cell wall biosynthesis and compromising the integrity of bacterial cell walls. Thanatin is a well-known AMP that effectively disrupts bacterial membranes, leading to cell death [110]. Additionally, the antimicrobial peptide Pse-T2 has demonstrated high-level, broad-spectrum antimicrobial activity and skin biocompatibility against multidrug-resistant *Pseudomonas aeruginosa* infections [111]. It exerts its effects by directly disrupting bacterial membranes. In biotechnological applications, plant defensins are highly attractive due to their low effective concentrations against fungi and their safety for mammals and birds. Transgenic plants expressing foreign defensins have shown enhanced resistance to various pathogens. AMPs also offer potential solutions to antibiotic resistance. For example, the antimicrobial peptide CM4 has been shown to combat *Pseudomonas aeruginosa* by both disrupting bacterial membranes and interfering with intracellular targets [112].

### 3.2. Activation of Plant Immune Responses

Plant peptides are pivotal in activating the immune system upon pathogen recognition. When plants detect pathogen-associated molecular patterns (PAMPs), peptides interact with specific receptors on plant cells, initiating signal transduction pathways. These interactions often activate mitogen-activated protein kinase (MAPK) signaling cascades [113,114]. The activation of MAPK pathways subsequently triggers the production of ROS, which act as signaling molecules and directly damage pathogens, and leads to the expression of defense genes encoding pathogenesis-related proteins [115]. For example, tomato SlSolP12 induces defense gene expression (e.g., phytoalexin and cell wall reinforcement genes), enhancing resistance to fungi, bacteria, and viruses [116].

### 3.3. Peptides Interact with Defensive Signaling Pathways

Plant immunity-related peptides participate in complex crosstalk within defensive signaling pathways, which often involve both antagonistic and synergistic interactions. These peptides are integral to PTI and ETI; for instance, PIP1 amplifies PTI by interacting with receptor-like kinase 7 (RLK7) [53]. They also modulate hormone-mediated defense pathways, including SA, JA, ethylene, and auxin pathways, contributing to both antagonistic and synergistic interactions. For example, in tobacco, a specific peptide can enhance JA-mediated defense while inhibiting SA-mediated defense upon binding to its receptor, depending on peptide and PAMP levels [55]. Conversely, in interactions with beneficial rhizobacteria, plant peptides may simultaneously trigger both JA and SA pathways, leading to a more comprehensive defense response [1]. Specific peptide–hormone interactions include systemin activating JA biosynthesis via the SR160 receptor to induce defense genes [101]; phytosulfokine (PSK) interacting with the SA pathway to enhance resistance to *Xanthomonas oryzae* in rice [31] and with auxin signaling to improve defense against *Botrytis cinerea* in tomato [60]; and CLE42 delaying leaf senescence by antagonizing ethylene signaling [17]. Additionally, IDA/IDL peptides synergize with flg22 to induce immune genes, indicating crosstalk with classic immune pathways [32], while RALF peptides interact with the FERONIA receptor to modulate ROS signaling and cell wall integrity, integrating with defense responses [38]. Collectively, these interactions form a complex network that coordinates plant immune responses against various pathogens.

### 3.4. Differences Between Plant Immunopeptides and Other Peptides

Plant immunity-related peptides possess distinct structural, functional, and mechanistic traits, which can be compared with immunity-associated peptides from other kingdoms. Structurally, plant peptides involved in immunity often have specific features: cysteine-rich peptides (CRPs) like plant defensins, characterized by the cysteine-stabilized α-helix and β-sheet (CSαβ) motif formed by conserved cysteine residues and disulfide bonds [89]; post-translationally modified peptides (PTMs) such as phytosulfokine (PSK) with sulfated tyrosine residues and CLE peptides with a conserved 12–13 amino acid C-terminal domain, which require modifications for activation [12,15]. Functionally, plant peptides like systemin and plant elicitor peptides (Peps) act as damage-associated molecular patterns (DAMPs) or phytocytokines, binding to receptors (e.g., SR160 for systemin, PEPR1/2 for Peps) to trigger downstream signaling [47,101]; PSK peptides balance defense and growth by suppressing bacterial immunity while enhancing resistance to necrotrophic pathogens [28]. Mechanistically, plant peptides rely on receptor-mediated signaling (e.g., CLE41/44 with BAM1) and integrate with hormone pathways [1,15]. Evolutionarily, plant peptides like defensins are conserved across land plants [89]. In contrast, peptides from other kingdoms differ in structural patterns, functional focuses, and action mechanisms, but all kingdoms utilize peptide-based immunity, highlighting the universal role of peptides in defense.

## 4. Applications of Plant Peptides in Disease Resistance

In the rapidly evolving field of plant pathology, recent research has highlighted the central role of plant peptides in enhancing disease resistance, with breakthroughs spanning from molecular mechanisms to agricultural translations. These short amino acid chains have emerged as versatile regulators, integrating into both direct pathogen inhibition and systemic immune signaling networks.

AMPs exhibit substantial potential for conferring broad-spectrum pathogen protection. For example, a plant-derived AMP-based biopesticide demonstrated comparable efficacy to chemical pesticides in controlling rice bacterial blight [117]. In citrus, a synthetic cecropin variant significantly suppressed *Candidatus Liberibacter asiaticus*—the causal agent of huanglongbing (HLB) [118]. A recent study employed AI-driven screening to identify anti-proteolysis peptides, such as APP3-14, which stabilizes MYC2 by inhibiting PUB21 activity. This peptide not only controlled HLB pathogen titers but also disrupted disease transmission, achieving up to 80% control efficiency in single-season trials [119]. These advancements establish plant peptides as modular components for precision disease management. Future research is anticipated to focus on engineering peptide–receptor interactions for crop-specific resistance and developing multi-mechanistic peptide cocktails, offering sustainable solutions for global food security.

## 5. Conclusions and Perspectives

Plant peptides, as key signaling molecules in the plant defense system, exhibit diverse and efficient regulatory mechanisms against pathogenic microorganism invasion. In terms of structural classification, whether post-translationally modified small peptides such as CLE and CEP, cysteine-rich peptides like RALF and defensins, or unmodified peptides such as systemin and Peps, they all function by directly inhibiting pathogens (disrupting pathogen cell membranes or inhibiting cell wall synthesis) or indirectly activating immune responses (inducing ROS bursts or regulating hormone signaling pathways). For example, CLE peptides inhibit nematode spread by interacting with receptor kinases, PSK peptides balance disease resistance and growth through dual regulatory mechanisms, and AMP-like peptides directly kill pathogens with broad-spectrum antimicrobial activity. Furthermore, the mechanisms of plant peptides show network-like regulation, not only integrating hormone pathways such as SA and JA but also triggering systemic immunity through receptor-mediated signal transduction. These findings provide new perspectives for understanding plant–pathogen interactions and lay a theoretical foundation for crop disease resistance improvement.

Future work should clarify non-classical peptide–receptor signaling and use CRISPR to engineer peptide expression for enhanced resistance. Meanwhile, the development of peptide-based biopesticides, such as synthetic protease-resistant peptide analogs and peptide–nanomaterial complexes, should address production cost issues via microbial fermentation or plant bioreactors. Multidisciplinary approaches integrating bioinformatics, structural biology, and synthetic biology are essential for functional exploration and application. Field studies on ecological impacts should also be conducted to ensure safety. Furthermore, synergistic systems combining plant peptides with RNA interference or microbiome regulation should be established to address global agricultural challenges like pathogen evolution under climate change, providing sustainable solutions for food security.

## Figures and Tables

**Figure 1 plants-14-02452-f001:**
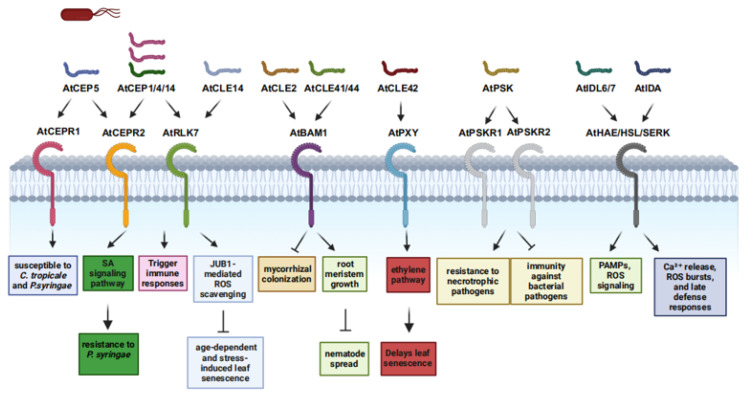
Functions of posttranslational modification peptides in plant immunity and their corresponding receptors. *C*. *tofeldiae: Colletotrichum tofeldiae*, *P*. *syringae: Pseudomonas syringae*.

**Figure 2 plants-14-02452-f002:**
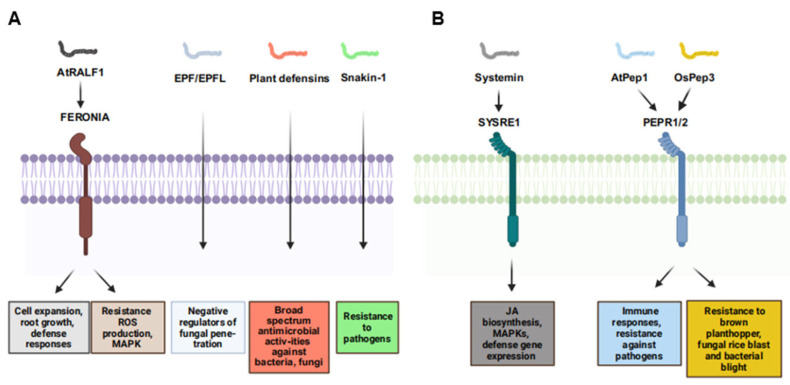
Functions of cysteine-rich and non-cysteine-rich/non-PTM peptides in plant immunity and their corresponding receptors. (**A**) Functions of cysteine-rich peptides in plant immunity and their corresponding receptors. (**B**) Functions of non-cysteine-rich/non-PTM peptides in plant immunity and their corresponding receptors.

## Data Availability

There are no new data associated with this article.

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
