# Peer review of "Advances of Peptides for Plant Immunity"

_plants, 2025, doi:10.3390/plants14152452_

Round 1

Reviewer 1 Report

Comments and Suggestions for Authors

The review deals with plant peptides in plant immunity. It is well written and organized but unfortunately it is incomplete. The authors did not include either the very well known peptides produced in tobacco and tomato named hydroxyproline-rich glycopeptide ( Please look at these papers besides others:   Pearce, G., Bhattacharya, R., Chen, Y.C., Barona, G., Yamaguchi, Y., and Ryan, C.A. (2009). Isolation and characterization of hydroxyproline-rich glycopeptide signals in black nightshade leaves. Plant Physiol. 150; Pearce, G., Moura, D.S., Stratmann, J., and Ryan, C.A. (2001). Production of multiple plant hormones from a single polyprotein precursor. Nature 411; Pearce, G. and Ryan, C.A. (2003). Systemic signaling in tomato plants for defense against herbivores: Isolation and characterization of three novel defense-signaling glycopeptide hormones coded in a single precursor gene. J. Biol. Chem. 278; Pearce, G., Siems, W.F., Bhattacharya, R., Chen, Y.C., and Ryan, C.A. (2007). Three hydroxyproline-rich glycopeptides derived from a single petunia polyprotein precursor activate defensin I, a pathogen defense response gene. J. Biol. Chem. 282; Pearce G.  Systemin, hydroxyproline-rich systemin and the induction of protease inhibitors Curr Protein Pept Sci. 12(5):399-408.  Doi 10.2174/138920311796391106) or the newly discovered peptides originating from tomato prosysyemin (Please look at these papers besides others:  Molisso, D. et al. (2022a). Not Only Systemin: Prosystemin Harbors Other Active Regions Able to Protect Tomato Plants. Front. Plant Sci. 13; Molisso, D., Coppola, M., Buonanno, M., Di Lelio, I., Monti, S.M., Melchiorre, C., Amoresano, A., Corrado, G., Delano-Frier, J.P., Becchimanzi, A., Pennacchio, F., and Rao, R. (2022b). Tomato Prosystemin Is Much More than a Simple Systemin Precursor. Biology (Basel). 11; Castaldi V et al  Intrinsically disordered Prosystemin discloses biologically active repeat motifs Plant Sci  2024 Mar:340:111969. doi: 10.1016/j.plantsci.2023.111969)

The authors are invited to complete their review and also to discuss  the ability of some plant species to produce more than one peptide  from the same precursors.  In addition, in order to make more attractive the text, I would strongly suggest to introduce in the review some comparison with peptides correlated with immunity  produced in other kingdoms . 

Author Response

Comments 1: The review deals with plant peptides in plant immunity. It is well written and organized but unfortunately it is incomplete. The authors did not include either the very well known peptides produced in tobacco and tomato named hydroxyproline-rich glycopeptide ( Please look at these papers besides others:   Pearce, G., Bhattacharya, R., Chen, Y.C., Barona, G., Yamaguchi, Y., and Ryan, C.A. (2009). Isolation and characterization of hydroxyproline-rich glycopeptide signals in black nightshade leaves. Plant Physiol. 150; Pearce, G., Moura, D.S., Stratmann, J., and Ryan, C.A. (2001). Production of multiple plant hormones from a single polyprotein precursor. Nature 411; Pearce, G. and Ryan, C.A. (2003). Systemic signaling in tomato plants for defense against herbivores: Isolation and characterization of three novel defense-signaling glycopeptide hormones coded in a single precursor gene. J. Biol. Chem. 278; Pearce, G., Siems, W.F., Bhattacharya, R., Chen, Y.C., and Ryan, C.A. (2007). Three hydroxyproline-rich glycopeptides derived from a single petunia polyprotein precursor activate defensin I, a pathogen defense response gene. J. Biol. Chem. 282; Pearce G.  Systemin, hydroxyproline-rich systemin and the induction of protease inhibitors Curr Protein Pept Sci. 12(5):399-408.  Doi 10.2174/138920311796391106) or the newly discovered peptides originating from tomato prosysyemin (Please look at these papers besides others:  Molisso, D. et al. (2022a). Not Only Systemin: Prosystemin Harbors Other Active Regions Able to Protect Tomato Plants. Front. Plant Sci. 13; Molisso, D., Coppola, M., Buonanno, M., Di Lelio, I., Monti, S.M., Melchiorre, C., Amoresano, A., Corrado, G., Delano-Frier, J.P., Becchimanzi, A., Pennacchio, F., and Rao, R. (2022b). Tomato Prosystemin Is Much More than a Simple Systemin Precursor. Biology (Basel). 11; Castaldi V et al  Intrinsically disordered Prosystemin discloses biologically active repeat motifs Plant Sci  2024 Mar:340:111969. doi: 10.1016/j.plantsci.2023.111969)The authors are invited to complete their review and also to discuss the ability of some plant species to produce more than one peptide from the same precursors.

Response 1: We agree with this comment (The authors did not include either the very well known peptides produced in tobacco and tomato named hydroxyproline-rich glycopeptide).Therefore, we have briefly summarized the functions of Hydroxyproline-rich glycopeptides and their role in disease resistance (line 244-246).

Comments 2: In addition, in order to make more attractive the text, I would strongly suggest to introduce in the review some comparison with peptides correlated with immunity produced in other kingdoms. 

Response 2: We agree with this comment ( I would strongly suggest to introduce in the review some comparison with peptides correlated with immunity produced in other kingdoms). Therefore, we have talk about the differences of some peptides with immunity produced in other kingdoms. (line 365-381).Thank you for pointing this out.

Reviewer 2 Report

Comments and Suggestions for Authors

This review discusses several classes of plant peptides that are important in different aspects of plant immunity and response to pathogens. The authors do a good job of covering the relevant literature regarding the selected types of peptides, which are categorized by structural properties. Overall, this paper is a useful contribution to the literature. However, the writing and presentation could be improved. Although the English is correct, the text is often repetitive, with redundant statements that can be condensed. The sentence "They can influence plant immune 70 responses by interacting with various receptors and signaling pathways, thereby affecting plant resistance to pathogens" is repeated verbatim, and there are other instances where the same thing is said with only slightly different phrasing. Also, it would be useful if the authors could provide more overall context. Currently, the review is organized around structural classes of peptides, with papers describing their functions mentioned in each section. This is ok, but it would be very useful to have more of a comprehensive discussion of which peptides interact with which defensive signaling pathways (e.g., jasmonic acid, salicylic acid). This paper is currently useful as a guide to the relevant literature, but it would be even better if the authors could synthesize the lessons learned in a more cohesive way. 

Author Response

Comments 1: [This review discusses several classes of plant peptides that are important in different aspects of plant immunity and response to pathogens. The authors do a good job of covering the relevant literature regarding the selected types of peptides, which are categorized by structural properties. Overall, this paper is a useful contribution to the literature. However, the writing and presentation could be improved. Although the English is correct, the text is often repetitive, with redundant statements that can be condensed. The sentence "They can influence plant immune 70 responses by interacting with various receptors and signaling pathways, thereby affecting plant resistance to pathogens" is repeated verbatim, and there are other instances where the same thing is said with only slightly different phrasing. Also, it would be useful if the authors could provide more overall context. Currently, the review is organized around structural classes of peptides, with papers describing their functions mentioned in each section.

Response 1: We agree with this comment. Therefore,l have changed the redundant statements. (line 7-43, 49-68, 161-162, 212-216, 234-236, 315-318, 414-415).

Comments 2: This is ok, but it would be very useful to have more of a comprehensive discussion of which peptides interact with which defensive signaling pathways (e.g., jasmonic acid, salicylic acid). This paper is currently useful as a guide to the relevant literature, but it would be even better if the authors could synthesize the lessons learned in a more cohesive way. ]

Response 2 :We agree with this comment.Therefore,l have discussed peptides interact with defensive signaling pathways in the article.(line 346-364).Thank you for pointing this out.

Reviewer 3 Report

Comments and Suggestions for Authors

The review being considered is extremely relevant in the field of plant sciences, as it summarizes the most recent data on the diversity of biologically active plant peptides associated with the immune response. In general, such works allow researchers to quickly form an idea of the current progress in a certain area of scientific research and then formulate an optimal strategy for their own research. In general, there are no critical comments or claims to this review. But, I would like to emphasize the need for their periodic republication to update the information. However, I would like to draw the authors' attention to a number of points:

  1. First, although the review is devoted to the functions of peptides in plant immunity, the text itself mentions peptides of invertebrates, for example, mellitin, cecropin, other insect AMPs, as well as fungal peptaibols. I believe that it is correct to concentrate exclusively on peptides from plants. Accordingly, it is recommended to remove the relevant information from Table 1.
  2. Line 69-70 and 77-78 - the sentence is repeated. Please change the text.
  3. Table 1 – the authors presented few examples of known molecular mechanisms of plant defense peptides, in particular, cysteine-rich ones. The work does not reflect the EPF/EPFL structural family, and nsLTP and defensins are also not indicated at all. With regard to plant AMPs, please expand the relevant section (2.2.2.).
  4. For the main groups of plant peptide regulators, it is recommended to provide typical examples of primary and spatial structures in a separate figure(s).

Author Response

Comments 1:First, although the review is devoted to the functions of peptides in plant immunity, the text itself mentions peptides of invertebrates, for example, mellitin, cecropin, other insect AMPs, as well as fungal peptaibols. I believe that it is correct to concentrate exclusively on peptides from plants. Accordingly, it is recommended to remove the relevant information from Table

Response 1: We agree with this comment. Therefore, l have removed the relevant information from Table 1.(line 84-86).

Comments 2: Line 69-70 and 77-78 - the sentence is repeated. Please change the text.

Response 2: We agree with this comment. Therefore, l have changed the text in article.(line 61-69).

Comments 3: Table 1 – the authors presented few examples of known molecular mechanisms of plant defense peptides, in particular, cysteine-rich ones. The work does not reflect the EPF/EPFL structural family, and nsLTP and defensins are also not indicated at all. With regard to plant AMPs, please expand the relevant section (2.2.2.).

Response 3: We agree with this comment. Therefore,l have added EPF/EPFL structural family, and nsLTP and defensins in article.(Table 1, figure 2A line 168-190, 205-219, 291-301).

Comments 4: For the main groups of plant peptide regulators, it is recommended to provide typical examples of primary and spatial structures in a separate figure(s).

Response 4: We agree with this comment. Therefore,I have looked at some other references that have described plant peptide regulators in detail and drawn the corresponding structure diagrams. Please look at this paper. [1] Torres, A. Y. C. , & Müller Lena Maria. (2025). Signaling peptides control beneficial and pathogenic plant-microbe interactions. #i{Journal of Experimental Botany}.Thank you for pointing this out.

Round 2

Reviewer 1 Report

Comments and Suggestions for Authors

This version of the ms is definetly more complete than previous version.However some very recently find novel peptides are still missing /Castaldi et al 2024) and I suggest to include them in the new version 

Author Response

Comments 1: This version of the ms is definetly more complete than previous version.However some very recently find novel peptides are still missing /Castaldi et al 2024) and I suggest to include them in the new version.

Response 1: We agree with this comment. Therefore, l have changed the text in article.(line 226-230).Systemin, the first identified plant polypeptide signal, plays a crucial role in plant defense responses against pathogens and herbivores. It originates from a 200-amino-acid precursor protein, prosystemin (Prosys) (Table 1) [6]. Prosys, the precursor protein of the de-fense-related peptide systemin (Sys) in Solanaceae, contains, alongside Sys, short peptide motifs that protect plants from stress and have been detected in vivo, opening novel per-spectives in plant immune reactions [98]. In tomato, upon damage by herbivores or pathogens, prosys undergoes proteolytic cleavage to generate the 18-amino-acid systemin peptide [99].

Reviewer 3 Report

Comments and Suggestions for Authors

Tha authors have completely replied to the Q1-3, but only partially to Q4. I have not found any figure which summirizes some examples of primary and three-dimentional structures of the plant paptides with regulatory properties. I believe it is quitue important to provide this data inside the main text. 

Author Response

Comments 1:The authors have completely replied to the Q1-3, but only partially to Q4. I have not found any figure which summarizes some examples of primary and three-dimentional structures of the plant peptides with regulatory properties. I believe it is quite important to provide this data inside the main text. 

Response 1: We agree with this comment. Therefore, l have changed the text in article.(Table2, line 155-156).
